# Importance of the Role of ω-3 and ω-6 Polyunsaturated Fatty Acids in the Progression of Brain Cancer

**DOI:** 10.3390/brainsci10060381

**Published:** 2020-06-17

**Authors:** Mayra Montecillo-Aguado, Belen Tirado-Rodriguez, Zhen Tong, Owen M. Vega, Mario Morales-Martínez, Shaheen Abkenari, José Pedraza-Chaverri, Sara Huerta-Yepez

**Affiliations:** 1Programa de Doctorado en Ciencias Biomédicas, Facultad de Medicina, Universidad Nacional Autónoma de Mexico (UNAM), Mexico City 04510, Mexico; mayramontecillo@hotmail.es; 2Hospital Infantil de Mexico, Federico Gomez, Unidad de Investigacion en Enfermedades Oncologicas, Mexico City 06720, Mexico; bely_16@hotmail.com (B.T.-R.); ixnergal@gmail.com (M.M.-M.); 3Molecular Toxicology Interdepartmental Program and Environmental Health Sciences, University of California, Los Angeles, CA 90095, USA; tong0817@g.ucla.edu; 4Department of Pathology & Laboratory Medicine, University of California, Los Angeles, CA 90095, USA; omvega@ucsd.edu (O.M.V.); shaheenabkenari@gmail.com (S.A.); 5Departamento de Biología, Facultad de Química, Universidad Nacional Autonoma de Mexico (UNAM), Mexico City 04510, Mexico; pedraza@unam.mx

**Keywords:** polyunsaturated fatty acids, brain cancer, oxylipins

## Abstract

Brain cancer is one of the most malignant types of cancer in both children and adults. Brain cancer patients tend to have a poor prognosis and a high rate of mortality. Additionally, 20–40% of all other types of cancer can develop brain metastasis. Numerous pieces of evidence suggest that omega-3-polyunsaturated fatty acids (ω-PUFAs) could potentially be used in the prevention and therapy of several types of cancer. PUFAs and oxylipins are fundamental in preserving physiological events in the nervous system; it is, therefore, necessary to maintain a certain ratio of ω-3 to ω-6 for normal nervous system function. Alterations in PUFAs signaling are involved in the development of various pathologies of the nervous system, including cancer. It is well established that an omega-6-polyunsaturated fatty acid (ω-6 PUFA)-rich diet has a pro-tumoral effect, whereas the consumption of an ω-3 rich diet has an anti-tumoral effect. This review aims to offer a better understanding of brain cancer and PUFAs and to discuss the role and impact of PUFAs on the development of different types of brain cancer. Considering the difficulty of antitumor drugs in crossing the blood–brain barrier, the therapeutic role of ω-3/ω-6 PUFAs against brain cancer would be a good alternative to consider. We highlight our current understanding of the role of PUFAs and its metabolites (oxylipins) in different brain tumors, proliferation, apoptosis, invasion, angiogenesis, and immunosuppression by focusing on recent research in vitro and in vivo.

## 1. Overview Brain Cancer

Primary brain tumors include a heterogeneous group of non-malignant and malignant tumors arising from the brain parenchyma and its related structures. These tumors are considered an important cause of high mortality in adults and children [1]. In 2018, the World Health Organization (WHO) reported over 18,078,957 new cases and 9,555,027 deaths for both men and women [2]. According to the Central Brain Tumor Registry of the United States (CBTRUS) Statistical Report in 2011–2015, new primary brain tumor diagnoses included 69.1% benign tumors (271,705 cases) and 30.9% malignant tumors (121,277 cases) in the USA [3]. The most frequently reported brain cancer was meningioma at 37.1%, pituitary cancer at 16.4%, glioblastoma at 14.7%, and nerve sheath at 8.5% [3]. The WHO classifies brain tumors according to risk factor, present symptoms, treatments, outcomes, and others [2,4,5]. Within brain tumors, meningioma, pituitary tumors, and low-grade astrocytoma are considered benign brain tumors, in which tumor cells rarely invade the normal cells around them. Benign tumor cells have distinct borders and a slow growing speed. In contrast, oligodendroglia and high-grade astrocytoma are malignant brain tumors; they quickly spread within the brain or spinal cord and have indistinct borders and rapid progression rates [5,6]. The symptoms that the patient with brain cancer presents depend on the specific site of the primary tumor in the brain [7]. In a study with malignant glioma, the patients experienced symptoms like headaches, memory loss, cognitive changes, motor deficits, language deficits, seizures, personality changes, visual problems, changes in consciousness, nausea or vomiting, sensory deficits, and papilledema [8,9]. Genetic-associated diseases are one of the risk factors for brain cancer, including Cowden disease, Gorlin syndrome, Li-Fraumeni syndrome, Neurofibromatosis types 1 and 2, Tuberous sclerosis complex, Turcot syndrome, and Von Hippel-Lindau disease. The administration of drugs such as sleeping pills and anti-histaminic drugs is another risk factor causing brain cancer [10]. Other than genetics and drugs, environmental factors like radiation (ionizing and non-ionizing) [10,11] or smoking can also cause brain cancer. Besides this, a previous medical history of epilepsy, head trauma, seizures, age, infectious agents, hormones, allergies, immune-related conditions, chronic inflammation, obesity, and nutritional factors potentially cause brain cancer as well [4,10,12]. Additionally, roughly 20–40% of all other cancers will eventually develop brain metastasis [13]. Moreover, the average survival for patients with brain metastases is typically less than 6 months [14]. In adults, the most frequent primary tumor types that can metastasize to the brain are lung and breast carcinomas and malignant melanoma (up to 75% of brain metastasis) [15,16]. In contrast, the primary tumors most likely to metastasize to the brain in children and teenagers are sarcomas and germ cell tumors [17]. Brain tumor treatments are limited due to the location of the organ. Standard treatments for several types of brain cancer include surgery, radiotherapy, and the administration of chemotherapy drugs [18]. Surgery is often affected by the tumor size and location [19]. Additionally, the administration of radiation is limited by the potential damage to surrounding tissue, which can lead to severe functional disabilities in patients [20,21].

Additionally, antitumor drugs are ineffective in the brain as they cannot cross the blood–brain barrier (BBB). The BBB is a semipermeable barrier that inhibits the diffusion of large molecules and hydrophilic molecules from the bloodstream to the cerebrospinal fluid. Most treatments will not be able to cross the barrier, and those that do require a large dosage to cross the barrier, which can lead to systemic toxicity [22,23,24,25]. All these factors make brain cancers the most difficult to treat, making it necessary to develop new therapeutic strategies. Polyunsaturated fatty acid (PUFAs) could be a potential therapeutic alternative because, in addition to being important components of the Central Nervous System (CNS), there are different studies that support the involvement of PUFAs in limiting the development of cancer. The objective of this study is to discuss the compiled evidence using a broad search of the related literature that demonstrates the effect and impact that ω-3 and ω-6 have on the advancement of different types of brain cancer.

## 2. Polyunsaturated Fatty Acids (PUFAs)

There are three types of naturally occurring fatty acids classified by the number of carbon–carbon double bonds present in their fatty acid side chains: saturated, monounsaturated, and polyunsaturated. In particular, PUFAs are fatty acids that contain 14 to 20 or more carbon atoms with several double bonds. PUFAs are important in the composition of the phospholipids of all cell membranes [26]. In 1929, Burr et al. showed that the polyunsaturated linoleic and linolenic fatty acids are essential dietary components [27,28]. PUFAs are classified as either ω-3 or ω-6 fatty acids depending on the position of the first unsaturated bond from the terminal methyl group. ω-3 PUFAs include α-linolenic acid (ALA) (18:3n-3), Eicosapentaenoic acid (EPA) (20:5n-3), and Docosahexaenoic acid (DHA) (22:6n-3), while ω-6 PUFAs include mostly linoleic acid (LA) (18:2n-6), γ-linolenic acid (GLA) (18:2n-6), and araquidonic acid (ARA) (20:4n-6). LA is the most abundant polyunsaturated fatty acid in nature, and it can be found mainly in seed oils, including soybean, sunflower, safflower, and in wheat germ, grape, hemp, corn, and cotton. ALA is found at high levels in leafy green vegetables, flaxseeds, walnuts, canola oil, and microalgae oil [29,30,31]. LA can be indirectly converted to ARA in the body and is the main PUFA in the Western diet, comprising more than 85% of the PUFA intake [32]. ALA is converted in vivo to EPA and DHA [33]. EPA and DHA are found in seafood, and in particular salmon, tuna, sardine, mackerel, and krill oils [34].

PUFAs, by oxidative reactions, produce oxylipins; these oxylipins are essential metabolites in our body. Three pathways can form the oxylipins: (1) the cyclooxygenase pathway (COX), (2) the lipoxygenase pathway (LOX), and (3) the cytochrome P450 pathway (CYP450). The metabolism of ARA through the COX pathway generates 2-series prostaglandins (PGE_2_) and thromboxanes (TXA_2_) [35]. Meanwhile, ARA metabolism through the LOX pathways generates hydroperoxyl-eicosatetraenoic acids (HpETE), which are quickly converted to hydroxy-eicosatetraenoic fatty acids (5- 12- and 15-HETE) employing glutathione peroxidase. The 4-series leukotrienes (LTB_4_, LTC_4_, LTD_4_) are generated at the same time [36]. Additionally, 4-series leukotrienes (Leuk 4) can also be generated from 5-HpETE by the LOX pathway [37]. The third pathway involves cytochrome 450 enzymes, which may have ω-hydroxylase or epoxygenase activity. The metabolism of ARA by ω-hydroxylase results in the formation of 19- and 20-HETE, while the activity of the epoxygenase leads to four epoxy-eicosatrienoic cis-acids (EETs), which can be converted to dihydroxy eicosatetraenoic acids (DiHETs) by the enzyme epoxy-hydroxylase (sEH). EETs have various biological activities, while DHET metabolites are considered to be biologically inactive [38]. Besides, EPA is the precursor of the 3-series prostaglandins (PGE_3_, PGF_3_), thromboxanes (TXA_3_), resolvins, protectins, and maresins through the COX pathway. Meanwhile, the LOX pathway produces the 5-series leukotrienes (LTB_5_, LTC_5_, LTD_5_). The CYP450 pathway, through ω-hydroxylase, generates HEPEs, and epoxygenase activity generates EpETEs (Epoxyeicosatetraenoic acids), which can be converted to Dihydroxy eicosatetraenoic acids (DiHETEs) again through this pathway [39] (Figure 1).

The right amount of dietary PUFAs are generally considered to have beneficial health effects. However, ω-3 and ω-6 PUFAs have opposing effects on metabolic functions in our bodies. Eicosanoid derivatives from ARA (ω-6 PUFAs) are generally pro-inflammatory and are involved in various pathological processes such as atherosclerosis, bronchial asthma, and inflammatory bowel disease [40]. Eicosanoids derivatives from EPA (ω-3 PUFAs) are predominantly anti-inflammatory; can inhibit platelet aggregation; and are therapeutic in some clinical conditions, such as collagen vascular diseases, hypertension, diabetes mellitus, metabolic syndrome, psoriasis, eczema, atopic dermatitis, coronary heart disease (CHD), atherosclerosis, and cancer [39]. Thus, a balanced ratio of ω-3 and ω-6 PUFAs in the diet is necessary for health.

A recent meta-analysis indicated that processed meat consumption was associated with a higher risk of brain tumors [41], while intakes of vegetables and fruits [42] can reduce the risk. In addition, it has been shown that fish rich in ω-3 polyunsaturated fatty acids have been found to be associated with a lower risk of several types of cancer and are beneficial for brain development. Wei et al. found in a meta-analysis that fish intake might be associated with a lower risk of brain cancer [43]. In contrast, a diet rich in ω-6 polyunsaturated fatty acids can be associated with a high risk of brain cancer [44].

The recommended dietary ratio of ω-6/ω-3 PUFAs for health benefits is 1:1–2:1. However, the current Western population has a high consumption of ω-6 PUFAs (the foods rich in ω-6 PUFAs are seed oils, including soybean, sunflower, safflower; wheat germ; grape; hemp; corn; and cotton) [29,30], with the ratio of ω-6/ω-3 PUFAs at 15:1 to 16.7:1 [45]. Global statistics that quantify the global consumption of key dietary fats and oils by country, age, and sex from 1990 to 2010 reveal that in 2010, the global mean intake of seafood ω-3 fats was 163 mg/day, with significant regional variation (50 to 700 mg/day) and national variation (5 to 3886 mg/day). The highest intakes were observed in Iceland (1189 mg/day), Barbados (1178 mg/day), Japan (995 mg/day), the Maldives, the Seychelles, Denmark, Malaysia, South Korea, and Thailand [46]. In another global study from 1980 to 2016, Stark et al. reported that regions with high EPA (and DHA blood levels (>8%)) included East Asia, Scandinavia, and areas with indigenous populations or populations not fully adapted to Westernized food habits. Very low blood levels of EPA (<4%) were observed in North America, Central and South America, Europe, the Middle East, Southeast Asia, and Africa [47]. In the United States, the estimated per capita consumption of soybean oil (the main source of LA, ω-6 PUFA) increased 1000-fold from 1909 to 1999. The availability of LA increased from 2.79% to 7.21%, whereas the availability of ALA (ω-3 PUFA) increased from 0.39% to 0.72%. Similarly, the ratio of LA to ALA increased from 6.4 to 10.0 in the same century [48]. These studies suggest a low level of ω-3 PUFA consumption in most of the world; the unbalance of ω-3/ω-6 diets could potentially lead to health issues.

## 3. Importance of PUFAs in Brain Functioning and Brain Cancer

PUFAs and oxylipins are necessary for normal physiological events in the nervous system, and they are highly prevalent in the synapses and retina [49]. ARA and DHA are the major PUFAs in the rat brain, comprising 81–90% and 6–12%, respectively [50]. The brain has two main sources of DHA plasma pools: the nonesterified pool and the lysophosphatidylcholine (LPC) pool. The plasma non esterified-DHA (NE-DHA) is the main contributor to brain DHA [51,52]. Significantly, LA and EPA are highly represented in the cerebellum, while ARA and DHA are in the cerebrum [53]. ω-3 PUFAs play a fundamental role in maintaining the membrane integrity and fluidity, which is crucial for neurotransmitter binding, and signaling [54]. In vivo studies indicate that dietary PUFA supplementation can alter the PUFA composition in both normal brain tissue [55] and intracerebrally implanted tumor tissue [56]. In addition, several studies showed that the gut microbiota composition affects the gut–brain axis and has been associated with different behavioral, mood, and psychological disorders (depression, anxiety, and autism). Importantly, ω-3 PUFAs restore the bacterium phylum *Firmicutes/Bacteroidetes* ratio and increase the taxa *Lachnospiraceae* that leads to the high production of anti-inflammatory compounds like butyrate [57]. The ω-3/ω-6 ratio is therefore important in maintaining an appropriate level of biological membrane fluidity, which is in turn essential for the ion channel function, membrane receptor activity, and release of neurohormones [58].

Remarkably, PUFAs have the advantage of being able to cross the BBB by a non-fenestrated layer of cerebral microvascular endothelial cells (BMEC) through two ways [59]: (1) membrane-localized fatty acid transport proteins (FATP1 [60], FATP4 [51], fatty acid translocase (FAT)/CD36 [51], and Mfsd2a [61]) and (2) cytosolic localized fatty acid-binding proteins (FABP3 [62], FABP4 [62], and FABP5 [23,51,62,63]); this ligand–receptor union facilitates the brain fatty acid uptake and trafficking [64]. FATP1 [60] and FABP5 [63] are key players that regulate the brain uptake of NE-DHA [52]; Mfsd2a is the major contributor of LPC-DHA [61]. The FATP-4, CD36, and FABP5 receptors promote the permeability of the LA [51]; whether these receptors facilitate DHA transport across the BBB has not been investigated [63]. Furthermore, FABP3, FABP5, and FABP7 are localized in neurons, glial cells, and other brain parenchymal cells [65] and facilitate the brain uptake of ARA, DHA, and EPA [66,67]; saturated fatty acids [67,68]; and DHA [67,69], respectively. Additionally, the BBB contains EP1 and EP2 receptors that, through the union PGE_2_, increase the BBB permeability via an increase in the tyrosine phosphorylation of occluding [70,71]. Other proposed models involve the delivery of PUFAs by lipoproteins in esterified form, in which low-density lipoprotein receptors bind to lipoproteins. Lipoprotein lipases on the surface of the cerebral endothelial cells detach PUFAs by the hydrolysis of ester bonds [72]. Finally, PUFAs can cross the BBB by passive diffusion through a flip-flop mechanism. These mechanisms depend on local fluctuations in ion densities that form strong local electrical fields across the membrane bilayer, which forms water pores. Ion leakage will reduce the strength of the local field, but the pore will be able to remain open for a long time [52,72,73]. In Figure 2, we present the combination of the three proposed models that explain how PUFAs cross the BBB.

The BBB is a semipermeable barrier that inhibits the diffusion of large molecules and hydrophilic molecules from the bloodstream to the cerebrospinal fluid. Most treatments will not be able to cross the barrier and those that do require a large dosage to cross the barrier, which can lead to systemic toxicity. Regarding drugs that can cross the BBB, most must cross by transmembrane diffusion, which favors high lipid solubility. However, after crossing the barrier the molecule must then travel through the liquid setting of the colony stimulating factor (CSF). A higher lipid solubility of the drug will cause lower amounts of the drug to get into the brain. Other methods of crossing are those of L-DOPA and caffeine, which use saturable transport systems [74].

Interestingly, a recent study shows that FABP5 can uptake fatty acids but not drugs like diazepam, pioglitazone, and troglitazone [23], which can cause treatment failure in neurological diseases.

In another hand, alterations in PUFA signaling are involved in the development of several diseases of the nervous system. Epidemiological studies suggest that systemic inflammation increases the risk of developing neurodegenerative conditions, including depression, schizophrenia, Alzheimer’s, and Parkinson’s disease, as well as cancer. Inflammatory processes derived from ω-6 PUFAs consumption induces increased cerebral oxidative damage and secondary neurotoxicity [75]; this could promote chronic neurodegeneration and neuroinflammation, which is a step in the potential development of cancer. Additionally, several reports associate the overexpression of enzymes such as Ciclooxigenase-2 (COX-2), prostaglandin E synthase (PGES), and Cytochrome P450 Family 2 and 4 (CYP2s and CYP4s) with the development of different types of brain cancer. The overexpression of these enzymes is proposed as a major factor in promoting tumor development, which is linked to increased aggressiveness and poor prognosis [76,77,78]. In particular, high-grade malignant gliomas with high mitotic levels present higher COX-2 expression than low-grade gliomas with less proliferative indexes. The higher tumoral expression of COX-2 is strongly correlated with poor and clinically more aggressive gliomas [79]. Below, we presented some examples of different studies on PUFAs and the effect of oxylipins on different processes that participate in the development of different types of brain cancer.

### 3.1. The Implication of PUFAs on Inflammation in Brain Cancer

Inflammation characterizes the course of acute and chronic diseases and is largely responsible for metabolic and behavioral changes in patients. Several studies indicate that during cancer, functional modifications take place, including an increased concentration of pro-inflammatory cytokines, which induce neuro-inflammation and promote tumor growth. PUFAs play a central part in modulating the inflammatory process in the development of brain cancer. It has been shown that the oxylipins derived from ω-3 PUFAs, such as LXA_4_, inhibit inflammatory pain processing through the inhibition of extracellular-regulated kinase (ERK) and c-Jun N-terminal kinase (JNK)-activated pathways in spinal astrocytes [80], and resolvin D1 inhibits the pro-inflammatory cytokine IL-1β expression in microglial cells [81]. In addition, the DHA derivate, 17*S*-hydroxy-containing docosanoids (17RS-HpDHA), and 17*S* series resolvins regulate both leukocytes to reduce infiltration in vivo and in glial cells by blocking their pro-inflammatory cytokine production [82].

In contrast, some studies suggest that the ω-6 PUFAs facilitate proinflammatory processes in brain cancer. Ferreira et al. evaluated tumor samples and showed that grade IV glioblastoma patients present high quantities of ARA and the pro-inflammatory prostanoids TXB_2_, PGD_2_, PGE_2_, and PGF_2α_ versus patients with grade II/III tumors. Interestingly, the patients with high levels of pro-inflammatory prostanoids had significantly decreased survival rates [83]. Importantly, PGE_2_ in glioblastoma cells increases the overexpression of immunosuppressive cytokines, such as interleukin 6 (IL-6), interleukin 10 (IL-10), and granulocyte-macrophage colony-stimulating factor (GM-CSF). Therefore, this process blocks T cell infiltration and proliferation and the subsequent suppression of the recruitment and proliferation of naive effector immune cells [84]. All this evidence shows us that PUFAs play a fundamental role in the regulation of inflammatory mediators that are known to affect the development of brain cancer.

### 3.2. Regulation of Proliferation in Brain Cancer Cells by PUFAs

One of the main characteristics of tumor cells is uncontrolled proliferation [85]. Several studies have shown the involvement of PUFAs in modulating this process in brain cancer progression. For example, ARA and DHA have been shown to modulate cell proliferation, differentiation, and migration through PKC activities in a fatty acid-binding protein (B-FABP)-dependent manner in glioblastoma multiforme cells (U87) [86]. Besides this and contrary to what has been reported, Leaver et al. showed that γ-linolenic acid (GLA) decreased proliferation both in vitro (C6 glioma cell line and multicellular glioma spheroids prepared from cell lines) and in vivo (implanted C6 glioma cells in rats) models through MTT assay and proliferating cell nuclear antigen (PCNA) and Ki-67 stain, respectively [87].

The oxylipins derived from ω-6 PUFAs play an important role in the promotion of the proliferation of brain cancer cells. Several studies highlight the participation of PGE_2_, which stimulates cell proliferation through binding its receptor prostaglandin E receptor subtype EP1 in the KMG4 glioma cell line [88]. PGE_2_ enhanced cell survival and proliferation through its ability to trans-activate the EGFR and to activate the β-catenin in an in vitro model with glioblastoma primary cultures [89]. It induces Id1 via the EP4-dependent activation of mitogen-activated protein kinase (MAPK) signaling, and the Egr1 transcription factor is important for the tumor cell self-renewal and radiation resistance [90]. Additionally, the exogenous administration of PGE_2_ also induced significant increases in cell proliferation in T98G human glioma cells [91]. Additionally, Payner et al. reported that microsomal prostaglandin E synthase-1 (mPGES-1) plays a critical role in promoting astroglioma cell growth via the PGE_2_-dependent activation of type II PKA (Protein kinase A) in the human astroglioma cell line U87-MG [76]. Furthermore, Ferreira et al. reported that the increase in PGD_2_ production in glioblastoma (GBM) patients correlates with an increase in the growth and invasion of glioma [83]. On the other hand, leukotrienes such as LTB_4_ and LXA_4_ regulate growth-related gene expressions, such as EGFR, cyclin E, p27, and caspase-8 in neuronal stem cells (NSCs) [92]. The exogenous administration of 20-HETE increases the growth of human glioma cells (U251) in vitro; transfected U251 cells with CYP4A1 complementary deoxyribonucleic acid (cDNA) increases its proliferation, and the implantation of these transfected cells in the brain of rats resulted in larger tumors compared with the controls [78].

These studies provide evidence that PUFAs affect the proliferative activities of tumors and may indicate potential brain tumor behavior. The evidence also strongly suggests that the inhibition of ω-6 PUFAs could have the potential to be a regulating strategy in tumor progression.

### 3.3. The Effect of PUFAs on Apoptosis in Brain Cancer

Apoptosis is a well-conserved and highly regulated mechanism of cell death for the removal of unnecessary, surplus, aged, or damaged cells. The dysregulation of apoptosis can result in the persistence of mutated cells, leading to malformations; autoimmune diseases; neurodegenerative diseases; and cancer, including brain cancer. Recently, PUFAs have been identified as important mediators for apoptosis modulation in brain tumors.

Treatment with ω-3 PUFAs, such as EPA, together with radiation increased apoptosis through the production of reactive oxygen species (ROS) in human C6 glioma cells [87]. The DHA-induced poly (ADP-ribose) polymerase (PARP) cleavage increased the population of sub-G1 cells and increased the number of terminal deoxynucleotidyl transferase dUTP nick end labeling (TUNEL)-positive cells, which are indicators of apoptosis, and caused an increase in autophagic activity in glioblastoma cells [93]. Additionally, EPA and DHA supplementation plus irradiation in rat astrocytoma cells (36B10) caused alterations in the fatty acid profile and enhanced radiation-induced cytotoxicity [94], which is mediated through the generation of free radicals and lipid peroxidation [95].

Unlike the evidence in other cancers, where ω-6 type PUFAs inhibit apoptosis [96], interestingly the reports on brain cancer indicate a contrary effect. For example, Bell et al. showed that GLA induces apoptosis in glioma spheroids derived from U87, U373, MOG-G-CCM, and C6 cell lines, which was evaluated through morphological and terminal deoxynucleotidyl transferase dUTP nick end labeling (TUNEL) assays [97]. Additionally, Leaver et al. demonstrated that the intra-tumoral administration of ARA and GLA using osmotic minipumps in an in vivo C6 glioma model augmented apoptosis levels. The apoptotic process was evaluated through morphological cell changes, the presence of phosphatidylserine cell surface, mitochondrial permeability, and TUNEL assay [87]. GLA-treated tumors have increased free radicals and lipid peroxides and a decreased antioxidant content, a decreased expression of oncogenes ras and B-cell lymphoma 2 (Bcl-2), and an enhanced activity of p53 [39]. Additionally, Lalier et al. has shown that the intracellular injection of PGE_2_ induced dose-dependent apoptosis in glioblastoma cells, which was dependent on the presence of pro-apoptotic protein, Bcl-2-associated X protein (Bax) [98]. There is accumulating evidence on the role of neuronal apoptosis in diseases of the nervous system. Researchers are now looking forward to the exciting prospect of developing effective therapeutic strategies based on the manipulation of this physiological process.

### 3.4. Induction of Angiogenesis and Metastasis by PUFAs and Its Implication in Brain Cancer Progression

Numerous studies have demonstrated that angiogenesis is up-regulated in cancer development and is necessary for tumor growth and metastasis in almost all types of cancers, including brain cancer [99,100]. Both PUFAs and the enzymes that participate in their metabolism have been shown in different studies to regulate angiogenic factors in brain cancer. For example, Wang et al. showed that the addition of DHA and etoposide resulted in the marked suppression of the expression of the FGF-2 and EGFR genes involved in angiogenesis in medulloblastoma cell lines (Daoy and D283) [101]. On the other hand, Xu et al. proposed that EGF EGFR binding led to the p38-MAPK activation pathway. This pathway induces Sp1/Sp3 transcription factors, and it seems necessary for the EGFR-dependent transactivation of the COX-2 promoter, increasing PGE_2_ production. In addition, they showed that the in vitro treatment of LN229/puro and SF767 glioma cells with PGE_2_ increased the vascular endothelial growth factor (VEGF) messenger ribonucleic acid (mRNA) expression. This pathway may contribute to the neovascularization of malignant gliomas [102]. Guo et al. demonstrated that the U251 human glioblastoma cancer cell line transfected with CYP4A1 cDNA (U251 O) increased the formation of 20-HETE, and this correlates with the high VEGF and pERK1/2 expression compared with the control U251 cells, which suggests that 20-HETE may have pro-angiogenic properties in U251 human gliomas [78]. Further, Chen et al. reported that the inhibition of 20-HETE with *N*-hydroxy-*N*′-(4-butyl-2-methylphenol) formamidine (HET0016) in an in vivo model (growth U251 human glioblastoma cancer cells in the rat cornea) reduces corneal neovascularization by 70% compared with a control vehicle. The author attributed this effect to the decreased VEGF, FGF-2 and EGF levels [103]. Additionally, the CYP2C11 enzyme in rat astrocytes and CYP4X1 in the rat brain formed EETs, which promoted blood vessel formation, angiogenesis, and vasodilation in the rat brain [77,104]. Moreover, Zhang et al. reported that all region isoforms (5,6-EET, 8,9-EET, 11,12-EET and 14,15-EET) EETs induce angiogenesis. However, 8,9-EET and 11,12-EET had a more potent activity. The authors suggest that the mechanism by which EETs induce endothelial mitogenic activity appears to involve the antagonism of the tyrosine kinase pathway [105]. The inhibition of EETs in brain tissue with 17-octadecenoic acid (17-ODYA) and miconazole significantly reduced the tumor capillary formation and glial tumor size and resulted in an increased animal survival time in an intracranial tumor rat model with RG2 cells [106].

Metastases can cause severe outcomes for CNS, and poor prognosis often leads to high mortality. However, the lack of statistics often leads to an underestimation of the incidence of even symptomatic brain metastases. Several studies showed the important role of PUFAs in the development of brain metastasis. Interestingly, DHA binds to and sequesters FABP7 (Fatty acid-binding protein 7) to the nucleus, resulting in a decreased cell migration in U87 MG cells [107]. A recent study by Zou et al. showed that astrocytes have a high content of PUFAs (mainly ARA, mead acid, and DHA) that act as “donors” of PPARγ in the invading cancer cells. In addition, the authors directly added AA into the culture medium and promoted the growth of brain metastatic cancer cells through the PPARγ pathway, enhancing cell proliferation and metastatic outgrowth in the brain [108]. In addition, DHA and EPA decreased the COX-2 mRNA expression and PGE_2_ production; it caused a decrease in migration in a matrigel invasion assay in the human melanoma cell line (70W) that metastasizes to the brain in mice. Additionally, the exposure to PGE_3_ significantly decreased invasion in an in vitro assay with 70W cells, a human melanoma cell line that metastasizes to the brain in nude mice [109].

Concerningly, ARA and tumor necrosis alpha (TNF-α)-modulated COX-2 expression in 70W cells generate an increase in the production of PGE_2_ and upregulate in vitro invasion [109]. The presence of PGE_2_ increased brain invasion and migration through several mechanisms. For example, Chiu et al. reported that 12-O-tetradecanoylphorbol-13-acetate (TPA) stimulation, through the activation of protein kinase C (PKC) and ERKs, increased the COX-2 gene expression; elevated PGE_2_ production; and promoted matrix metalloproteinase-9 (MMP-9) activation, which induced in vitro migration/invasion in U87 glioblastoma cells [110]. Another study by Wang et al. demonstrated that the nuclear factor of activated T cells 1 (NFATC1) promoted the induction of COX-2 and PGE_2_, causing glioblastoma U251 cell invasion [111]. Additionally, Wang showed that PGE_2_ increased the migration of human glioblastoma cell lines in vitro through the prostaglandin receptors, EP2 or EP4 [112]. Additionally, Gomes et al. reported that the addition of exogenous PGE_1_ and PGE_2_ in T98G human glioma cells causes an increase in cell migration compared with controls through a transwell migration assay [91].

The pathophysiology of brain metastases is a complex-multistage process mediated by molecular mechanisms. From the primary organ, cancer cells must transform, grow, and be transported to the CNS where they can lay dormant for a long time before invading and growing. Understanding the pathophysiology of brain metastasis is necessary, because it may lead to the development of more efficient therapies to combat brain tumor growth or to possibly making the CNS an undesirable environment for tumor progression. Treatment by PUFAs is a possible option to combat brain cancer with the advantage that PUFAs can easily cross the BBB. Figure 3 shows the participation of PUFAs and oxylipins in the development of brain cancer; ω-3 PUFAs have an anti-tumoral activity, while ω-6 PUFAs have a pro-tumoral activity.

## 4. Concluding Remarks and Future Perspectives

Brain cancer represents one of the most aggressive types of cancer, with a poor prognosis and a high mortality rate. The evidence that we present demonstrates that the consumption of PUFAs, particularly ω-3 PUFAs, may be a potential therapeutic alternative since they are differentially expressed in different parts of the brain and constitute a fundamental part in the development and neurological functioning of the CNS. In contrast, the consumption of w-6 PUFAs are implicated in the development of brain cancer. Interestingly, PUFAs can easily cross the BBB, unlike some chemotherapeutic drugs.

Several studies in different types of brain cancer, primarily glioma, have shown that the presence of ω-3 PUFAs/oxylipins downregulates tumor growth, proliferation, angiogenesis, and metastasis. Meanwhile, ω-6 PUFAs/oxylipins induce an increase in these pro-tumor processes. These data are relevant, considering that the human population is exposed to a higher proportion of ω-6 than ω-3 PUFAs in their diet. Another important aspect is that several of the enzymes involved in the metabolism of PUFAs, such as COX-2, CYP2J2, CYP4A11, and sEH, are involved in the accumulation of PUFAs that result in the increase in tumors, because there are highly expressed in various types of brain cancer; therefore, it is important to consider the use of inhibitors of these enzymes as an alternative treatment. Considering this, oxylipins and PUFAs are difficult to measure.

## Figures and Tables

**Figure 1 brainsci-10-00381-f001:**
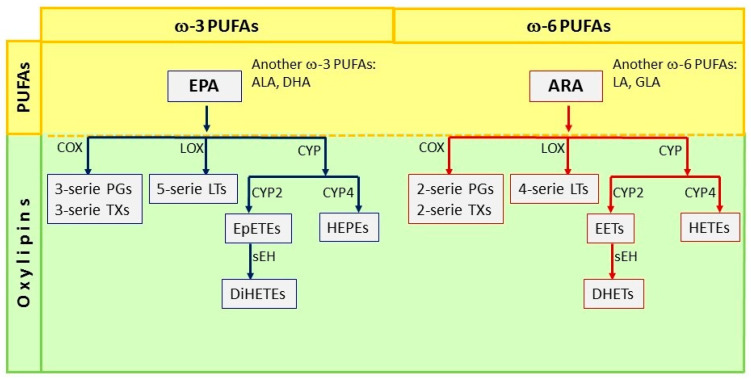
Pathways of synthesis of oxylipins from Eicosapentaenoic acid (EPA) and ARA PUFAs. ω-3 polyunsaturated fatty acids (PUFAs) include α-linolenic acid (ALA), Eicosapentaenoic acid (EPA), and Docosahexaenoic acid (DHA); the metabolism of EPA by Cyclooxygenase (COX) leads to the formation of 3-series prostaglandins (PGs) and 3-series Thromboxanes (TXs); the Lipoxygenase (LOX) pathway generates 5-series Leukotrienes (LTs); and the cytochrome P450 (CYP2) epoxygenases leads to the formation of Epoxyeicosatetraenoic acids (EpETEs). EpETEs are further metabolized by soluble epoxide hydrolase (sEH) to form the fatty acid diols termed Dihydroxy eicosatetraenoic acids (DiHETEs); CYP4 generates hydroxyeicosapentaenoic acids (HEPEs). In contrast, ω-6 PUFAs include linoleic acid (LA), γ-linolenic acid (GLA), and araquidonic acid (ARA). The metabolism of ARA through the COX pathway generates 2-series prostaglandins (PGs) and thromboxanes (TXs), while through the LOX pathway it generates the 4-series leukotrienes; ω-hydroxylase results in the formation of hydroxyeicosatetraenoic acids (HETEs); the activity of the epoxygenase leads to epoxy-eicosatrienoic acids (EETs), which can be converted to dihydroxy eicosatetraenoic acids (DiHETs) by the enzyme epoxy-hydroxylase (sEH).

**Figure 2 brainsci-10-00381-f002:**
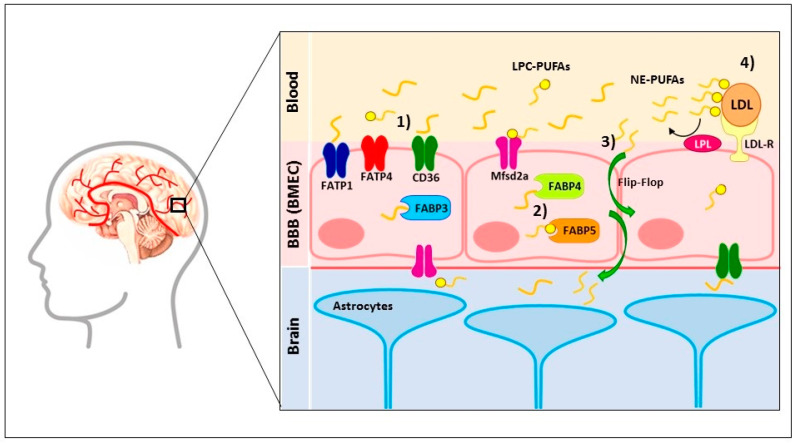
Mechanisms that polyunsaturated fatty acids (PUFAs) cross the blood–brain barrier (BBB). PUFAs have the advantage of being able to cross the BBB by the non-fenestrated layer of cerebral microvascular endothelial cells (BMEC). Four ways are proposed: (1) membrane-localized fatty acid transport proteins (FATP1, FATP4, fatty acid translocase (FAT)/CD36, and Mfsd2a). (2) Cytosolic-localized fatty acid binding proteins (FABP3, FABP4, and FABP5); this ligand–receptor union facilitates the brain fatty acid uptake and trafficking. FATP1 and FABP5 are key players that regulate the brain uptake of NE-DHA. Mfsd2a is the major contributor of LPC-DHA; the FATP-4, CD36, and FABP5 receptors promote the permeability of the LA. (3) PUFAs can cross the BBB by passive diffusion through a flip-flop mechanism. (4) Low-density lipoproteins (LDL) bind to the low-density lipoprotein receptor (LDL-R), and then lipoprotein lipases (LPL) liberate PUFAs by the hydrolysis of ester bonds. LPC-PUFAs—Lysophosphatidylcholine-Polyunsaturated fatty acids, NE-PUFAs—Non-Esterified-Polyunsaturated fatty acids.

**Figure 3 brainsci-10-00381-f003:**
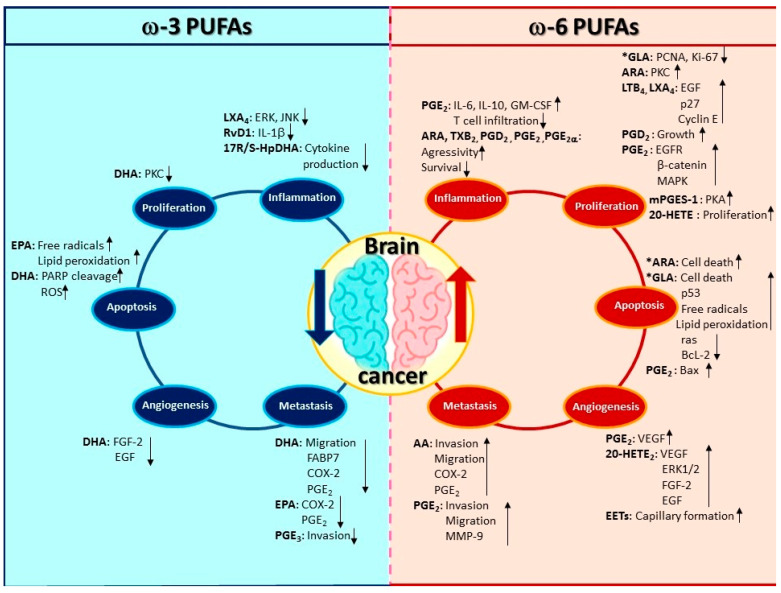
PUFAs and oxylipins involved in brain cancer development. PUFAs play a central part in modulating the different mechanisms involved in the development of brain cancer. In general, the evidence shows that ω-3 PUFAs have antitumor effects, whereas ω-6 PUFAs are attributed a pro-tumor tumor effect. However, interestingly GLA has antiproliferative effects and ARA and GLA show proapoptotic effects. Polyunsaturated fatty acids (PUFAs), Eicosapentaenoic acid (EPA), α-linolenic acid (ALA), Docosahexaenoic acid (DHA), Arachidonic acid (ARA), Linoleic acid (LA), γ-linolenic acid (GLA), Resolvin D1 (RvD1), 17*S*-hydroxy-containing docosanoids (17RS-HpDHA), Hydroxy-eicosapentaenoic acids (HEPEs), epoxy-eicosatrienoic cis-acids (EETs), 20-Hydroxy-eicosatetraenoic acids (20-HETE), Interferon-gamma (IFN-γ), Interleukin 6 (IL-6), Interleukin 10 (IL-10), granulocyte-macrophage colony-stimulating factor (GM-CSF), protein kinase C (PKC), Mitogen-Activated Protein Kinases (MAPK), Prostaglandin D_2_ (PGD_2_), Prostaglandin E_1_ (PGE_1_), Prostaglandin E_2_ (PGE_2_), Prostaglandin E_2α_ (PGE_2α_), Prostaglandin E_3_ (PGE_3_), Leukotriene B_4_ (LTB_4_), Lipoxin A_4_ (LXA_4_), poly (ADP-ribose) polymerase (PARP), Vascular Endothelial Growth Factor (VEGF), fatty acid-binding protein 7 (FABP7), extracellular signal-regulated kinase (ERKs), Epidermal growth factor receptor (EGFR), and Matrix metalloproteinase-9 (MMP-9).

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
