# Peer review of "Importance of the Role of ω-3 and ω-6 Polyunsaturated Fatty Acids in the Progression of Brain Cancer"

_brainsci, 2020, doi:10.3390/brainsci10060381_

Round 1

Reviewer 1 Report

General comment

When I was asked to review the manuscript I was attracted to the title. The word “Implication” could have several meanings in this context, but the inclusion of the word “development” was felt suggestive that the PUFA’s would be somewhere involved or at least related to the development of brain cancer. The text further did not meet my expectations. Hence the title should be adapted to the content that is provided.

Still the topic discussed in the review has high value.

I would suggest the authors to make a clear review question and to try to generate an answer. The goal of the study is now described at lines 83-85. This and what follows gives me more the impression of the introduction of a PhD thesis rather than a review study with a specific study question. Going the same line, one should provide in a review a literature search strategy. Finally one should give in the conclusion an answer to the question studied.

I think one should keep clear the terms “association” versus “causality”.

I think that the text can be restructured more according to the Preamble for IARC monographs on the identification of carcinogenic hazards to humans (https://monographs.iarc.fr/wp-content/uploads/2019/07/Preamble-2019.pdf): the exposure characterization, studies of cancer in humans, studies of cancer in experimental animals (and cells), and mechanistic evidence. If the authors succeed in moving the text at least into this direction, the work will be of much more value.

Some more specific comments

Line 59. The statement of the sleeping pills and anti-histaminic drugs is a strong statement with community-wide impact and should require a reference.

Around line 61. Curiously non-ionizing radiation is not mentioned.

Line 85. Is reference 25 appropriate ?

Line 149-166. The reader asks here automatically if there is a difference in the incidence and prevalence of brain cancer in the different countries having different intake of PUFA’s. The evidence for this should be assessed.

Line 253. “promote” suggests causality, but is it really “promote” or rather “facilitate”, or “is connected to” ?

Line 295-296. Is the conclusion not too strong ?

Author Response

Answer to the Reviewer Comments Manuscript ID: brainsci-811403

Comments to the Reviewer 1

Question 1. When I was asked to review the manuscript, I was attracted to the title. The word “Implication” could have several meanings in this context, but the inclusion of the word “development” was felt suggestive that the PUFA’s would be somewhere involved or at least related to the development of brain cancer. The text further did not meet my expectations. Hence the title should be adapted to the content that is provided.

Answer 1. Thank you for your comments. They will be very helpful for the improvement of this manuscript. You are correct, “Implication” is a very general word here and “Development” does not fully convey what we want to say on the manuscript. We have changed the title to “Importance of the role of ω-3 and ω-6 polyunsaturated fatty acids in the progression of brain cancer”.

Question 2. I would suggest the authors to make a clear review question and to try to generate an answer. The goal of the study is now described at lines 83-85. This and what follows gives me more the impression of the introduction of a PhD thesis rather than a review study with a specific study question. Going the same line, one should provide in a review a literature search strategy. Finally one should give in the conclusion an answer to the question studied.

Answer 2. We have changed the wording of the selected part. We have also included in the conclusion a connection to the question as to be presented as an answer.

Question 3. I think one should keep clear the terms “association” versus “causality”.

Answer 3. We have removed the word of association on lines 52, 228, and 352

Question 4. I think that the text can be restructured more according to the Preamble for IARC monographs on the identification of carcinogenic hazards to humans (https://monographs.iarc.fr/wp-content/uploads/2019/07/Preamble-2019.pdf): the exposure characterization, studies of cancer in humans, studies of cancer in experimental animals (and cells), and mechanistic evidence. If the authors succeed in moving the text at least into this direction, the work will be of much more value.

Answer 4. Thank you very much for your suggestion. However, all co-authors have discussed and have considered that the format of the manuscripts is the most appropriate way we want to present the information.

Question 5. Line 59. The statement of the sleeping pills and anti-histaminic drugs is a strong statement with community-wide impact and should require a reference.

Answer 5. The reference has been included (line 60).

Question 6. Around line 61. Curiously non-ionizing radiation is not mentioned.

Answer 6. We have included both ionizing and non-ionizing radiation (line 62)

Question 7. Line 85. Is reference 25 appropriate?

Answer 7. We have moved reference 25 further above where we believe it is more appropriate.

Question 8. Line 149-166. The reader asks here automatically if there is a difference in the incidence and prevalence of brain cancer in the different countries having different intake of PUFA’s. The evidence for this should be assessed.

Answer 8. We include the evidence in the new version of the manuscript.

Question 9. Line 253. “promote” suggests causality, but is it really “promote” or rather “facilitate”, or “is connected to”?

Answer 9. We have changed the word from “promote” to “facilitate” line 269.

Question 10. Line 295-296. Is the conclusion not too strong?

Answer 10. We have changed the wording of this paragraph as to not imply such a strong conclusion (lines 311-312).

Reviewer 2 Report

The authors present the current knowledge on Omega3 and 6 fatty acids in the development of bran cancer. 

As a neurosurgeon and clinician I cannot evaluate the whole manuscript as its content is out of my field. 

The introduction needs extensive revision. 

Brain tumors cover a wide spectrum of different tumor entities (see WHO brain tumor classification), including risk factors, presenting symptoms, treatment, outcome, ...

The authors generalize all information they present on all types of brain tumors . They should focus on certain entities (as they did when discussing the role of PUVAS: malignant glioma and metastasis). For me it makes no sense to compare e.g. glioblastoma with pituitary adenomas. 

The authors state that there are several risk factors for brain cancer, however there is no evidence for these factors.

Author Response

Answer to the Reviewer Comments Manuscript ID: brainsci-811403

Comments to the Reviewer 2

The authors present the current knowledge on Omega3 and 6 fatty acids in the development of brain cancer.  As a neurosurgeon and clinician, I cannot evaluate the whole manuscript as its content is out of my field. 

Question 1. The introduction needs extensive revision. 

Answer 1. Thank you very much for the suggestion. We have revised the introduction.

Question 2. Brain tumors cover a wide spectrum of different tumor entities (see WHO brain tumor classification), including risk factors, presenting symptoms, treatment, outcome, ...

Answer 2. In the new version of the manuscript we mentioned the WHO classification of brain tumors (lines 47-48)

Question 3. The authors generalize all information they present on all types of brain tumors. They should focus on certain entities (as they did when discussing the role of PUVAS: malignant glioma and metastasis). For me it makes no sense to compare e.g. glioblastoma with pituitary adenomas. 

Answer 3. Thank you for your comment. However, in this manuscript our intention is to give an overview of the effects of ω-3 and ω-6 PUFAs on different types of brain tumors, but we understand that malignant glioma is one of the most important.

Question 4. The authors state that there are several risk factors for brain cancer, however there is no evidence for these factors.

Answer 4. Evidence for these risk factors is presented in lines 61-63. These factors are mentioned in references 8-10.

Reviewer 3 Report

The review article on the implication of PUFA in the development of brain cancer is well written and comprehensively collected literature information. It includes roles various PUFA starting from different types of metabolites and their role inflammation, proliferation, angiogenesis, metastasis, and potential tools BBB. I have a few minor concerns in the review before recommending the article.

1)    Line 149-166  mentioned about the various composition of PUFA in diet across countries. Is there any study that showed a correlation or prevalence of low brain cancer rate in the population that consumes high omega-3 vs diet rich in omega-6?

2) Fig2 - I would recommend include index or label NE_PUFA & LPC-PUFA? They introduced the Flip-Flop mechanism in the image and line 197, would be appropriate if you could include a few sentences about the mechanism.

3) Line 337 should be " EGF to its EGFR" a minor correction.

4) The recent finding from Zhou et al.Cancer Discovery 2019 showed ARA from Astrocytes promotes Brain Metastatic cells. Recommend authors include this information in the Metastasis topic.

Author Response

Answer to the Reviewer Comments Manuscript ID: brainsci-811403

Comments to the Reviewer 3

The review article on the implication of PUFA in the development of brain cancer is well written and comprehensively collected literature information. It includes roles various PUFA starting from different types of metabolites and their role inflammation, proliferation, angiogenesis, metastasis, and potential tools BBB. I have a few minor concerns in the review before recommending the article.

Question 1. Line 149-166 mentioned about the various composition of PUFA in diet across countries. Is there any study that showed a correlation or prevalence of low brain cancer rate in the population that consumes high omega-3 vs diet rich in omega-6?

Answer 1. There is evidence of a correlation between low brain cancer rate and consumption of ω-3 rich diet vs ω-6 rich diet. We have included this in the new version of the manuscript (lines 150-155)

Question 2. Fig2 - I would recommend include index or label NE_PUFA & LPC-PUFA? They introduced the Flip-Flop mechanism in the image and line 197, would be appropriate if you could include a few sentences about the mechanism (lines 209-212)

Answer 2. We have included the modifications suggested for figure 2 and have also included a summary of flip-flop mechanism.

Question 3. Line 337 should be " EGF to its EGFR" a minor correction.

Answer 3. We have corrected this error.

Question 4. The recent finding from Zhou et al. Cancer Discovery 2019 showed ARA from Astrocytes promotes Brain Metastatic cells. Recommend authors include this information in the Metastasis topic.

Answer 4. We have included this paper in our manuscript (lines 379-383)

Reviewer 4 Report

  1. The authors presented a comprehensive review on the potential application of PUFAs against brain cancer.  They provided a good overview from basics to the latest especially on PUFA transport in the brain. However, the authors only had a limited segment on the role of PUFAs inflammation considering that one of the primary roles of PUFA is in inflammation and inflammation is a key process in the development of cancer. The manuscript can be improved by adding the citation of recent studies especially on the role of the microbiome and immune checkpoint inhibitors and possible interaction with PUFAs to brain cancer.

One of the recent developments on the use of ICI are the findings that its efficacy depends on the patients’ gut microbiome. Please refer to references below.

  • Gong, J et al (2018). Development of PD-1 and PD-L1 inhibitors as a form of cancer immunotherapy: a comprehensive review of registration trials and future considerations. J Immunother Cancer 6: 8, doi: 10.1186/s40425-018-0316-z.

It is worth citing the reference below showing the relationship between the microbiome and brain which bridges the microbiome and ICI in brain cancer

  • Sampson, J.H., Gunn, M.D., Fecci, P.E. et al. (2020) Brain immunology and immunotherapy in brain tumours. Nat Rev Cancer 20:12–25. https://doi.org/10.1038/s41568-019-0224-7

The articles below provide additional mechanisms on how PUFAs can regulate the microbiome and ICI which will strengthen the manuscript with additional biomarkers to consider for brain cancer and inflammation.

  • Constantini L et al (2017) Impact of Omega-3 Fatty Acids on the Gut Microbiota. Int J Mol Sci.2017 Dec 7;18(12). pii: E2645. doi: 10.3390/ijms18122645
  1. On the text, just a suggestion to include a reference on a botanical source for omega 3 PUFAs, such as seaweed or algae, which are the primary sources anyway.

Author Response

Answer to the Reviewer Comments Manuscript ID: brainsci-811403

Comments to the Reviewer 4

The authors presented a comprehensive review on the potential application of PUFAs against brain cancer.  They provided a good overview from basics to the latest especially on PUFA transport in the brain. However, the authors only had a limited segment on the role of PUFAs inflammation considering that one of the primary roles of PUFA is in inflammation and inflammation is a key process in the development of cancer. The manuscript can be improved by adding the citation of recent studies especially on the role of the microbiome and immune checkpoint inhibitors and possible interaction with PUFAs to brain cancer.

Question 1. One of the recent developments on the use of ICI are the findings that its efficacy depends on the patients’ gut microbiome. Please refer to references below.

-Gong, J et al (2018). Development of PD-1 and PD-L1 inhibitors as a form of cancer immunotherapy: a comprehensive review of registration trials and future considerations. J Immunother Cancer 6: 8, doi: 10.1186/s40425-018-0316-z.

Answer 1. Thank you very much for your suggestion. However, we do not believe that the referenced information fits well in our study.

Question 2. It is worth citing the reference below showing the relationship between the microbiome and brain which bridges the microbiome and ICI in brain cancer

-Sampson, J.H., Gunn, M.D., Fecci, P.E. et al. (2020) Brain immunology and immunotherapy in brain tumours. Nat Rev Cancer 20:12–25. https://doi.org/10.1038/s41568-019-0224-7

Answer 2. We extensively reviewed the mentioned reference, but we could not find the information about microbiome and brain which bridges the microbiome and ICI in brain cancer.

Question 3. The articles below provide additional mechanisms on how PUFAs can regulate the microbiome and ICI which will strengthen the manuscript with additional biomarkers to consider for brain cancer and inflammation.

-Constantini L et al (2017) Impact of Omega-3 Fatty Acids on the Gut Microbiota. Int J Mol Sci.2017 Dec 7;18(12). pii: E2645. doi: 10.3390/ijms18122645

Answer 3. We have added information from this reference in the new version of the manuscript. Lines 186-190

Question 4. On the text, just a suggestion to include a reference on a botanical source for omega 3 PUFAs, such as seaweed or algae, which are the primary sources anyway

Answer 4. In the new version of the manuscript, we included the reference of the source of the omega 3 PUFAs. Line 99
